# Gastrodin alleviates neuronal damage in epileptic cell models by targeting P2RY12 to inhibit microglial hyperactivation

**Aiyuan Cai[1], Zilong Li[1], Ran Liu[1], Hailong Huang[1], Ping Liu[1], Ruizhong Zhang[1], Jing Xiao[1], Yuanhong Lin[2], Qingpeng Hu**📍[1]*, **Haixia Wu[1]***

**1** Department of Pediatrics, Longhua District People's Hospital, Shenzhen, Guangdong, China, **2** Second Clinical Medical College, Guangzhou University of Chinese Medicine, Guangzhou, China

* huqingpeng163@126.com (QH); 13923430437@163.com (HW)

## Abstract

Epilepsy is a chronic neurological disorder characterized by recurrent seizures, with its onset and progression closely linked to neuroinflammation, where abnormal activation and migration of microglia serve as a key pathological process. This study focuses on the regulatory role of gastrodin, the principal active component of the traditional Chinese medicine Gastrodia elata, and its target P2RY12 in chronic epilepsy. Through bioinformatics analysis, P2RY12 was identified as a potential target for Gastrodia elata in treating epilepsy. Molecular docking, Pull-down, and cellular thermal shift assays confirmed that Gastrodin can directly bind to the P2RY12 receptor and inhibit the activation of its downstream RhoA/ROCK signaling pathway. In an *in vitro* epilepsy model induced by kainic acid (KA), Gastrodin intervention significantly suppressed the migration of microglia toward the injured area and reduced the rearrangement of the F-actin cytoskeleton. Meanwhile, Gastrodin markedly decreased the release of pro-inflammatory cytokines such as TNF-α and IL-1β, alleviated neuronal calcium overload, and inhibited cell apoptosis. Mechanistically, Gastrodin exerts its neuroprotective effects against epilepsy by targeting the P2RY12 receptor to inhibit its mediated chemotactic signaling pathway and inflammatory response, thereby reducing epilepsy-related neuronal damage. Notably, the protective effects of Gastrodin were further enhanced after P2RY12 expression was interfered with, further confirming the critical role of P2RY12 in its neuroprotective action. This study reveals, at the receptor-signaling axis level, the molecular mechanism by which Gastrodin regulates microglial function through P2RY12 to exert anti-epileptic effects, providing not only a solid scientific basis for the clinical application of the traditional Chinese medicine Gastrodia elata but also a novel potential strategy for the treatment of chronic epilepsy.

**Data availability statement:** All relevant data are within the manuscript and its Supporting Information files.

**Funding:** This research is funded by Protective effect of SAHA on brain injury in developing epileptic rats and regulation of histone acetylation of TLR4 gene (2023JJ30532); The anti-epileptic mechanism of xyloketal B was explored based on network pharmacology from SIRT1/NF-κB/GMD mediated astrocytic pyroptosis, Excellent Project of the Shenzhen Longhua District Science and Technology Innovation Bureau (2025012); Research on improving the teaching effect of pediatric neurological diseases in Longhua District People's Hospital, teaching reform of Shenzhen Longhua District People's Hospital (2024251). The funders had no role in study design, data collection and analysis, decision to publish, or preparation of the manuscript.

**Competing interests:** The authors have declared that no competing interests exist.

**Abbreviation:** ASMs: antiseizure medications; CNS: Central Nervous System; DAMPs: Damage-Associated Molecular Patterns; P2RY12: P2Y Purinoceptor 12; GPCR: G - protein - coupled receptor; RhoGEF: Rho guanine nucleotide exchange factor; ROCK: Rho - associated kinase; DEG: differentially expressed gene; GO: Gene Ontology; KEGG: Kyoto Encyclopedia of Genes and Genomes; KA: kainic acid; EM: Epilepsy Model; NES: normalized enrichment score; CETSA: cellular thermal shift assay; SYP: Synaptophysin; MAP2: Microtubule-Associated Protein 2.

## 1. Introduction

Epilepsy is a common chronic neurological disorder characterized by recurrent seizures, affecting over 50 million people worldwide. Its core pathological feature is recurrent, paroxysmal cerebral dysfunction caused by abnormal, highly synchronized neuronal discharges [1–4]. The clinical manifestations epileptic seizures are diverse, ranging from localized muscle twitches to generalized loss of consciousness [5]. Currently, the treatment of epilepsy mainly relies on antiseizure medications (ASMs). However, approximately one – third of patients respond poorly to existing drugs and are classified as having drug – refractory epilepsy. Moreover, the scope of surgical intervention is also very limited [6,7]. Traditional drugs mainly target ion channels or neurotransmitter systems. Although they can control symptoms in some patients, they are often unable to block the progressive damage and recurrence mechanisms of the disease [8,9]. In recent years, research has shown that the pathogenesis of epilepsy extends far beyond simple "electrical disorders" and involves multiple pathological processes, such as neuroinflammation, glial cell activation, blood – brain barrier disruption, and abnormal immune responses. Among these, neuroinflammation is widely recognized as a key driving factor in the onset and development of epilepsy [10–14].

Microglia, the main tissue – resident macrophage – like immune cells in the central nervous system (CNS), play a central role in the regulation of neuroinflammation [15–17]. Under physiological conditions, microglia exhibit a branched, resting state and continuously monitor the brain microenvironment [18]. After an epileptic seizure, the brain microenvironment undergoes dramatic changes. Large amounts of damage - associated molecular patterns (DAMPs), such as ATP, are released from damaged neurons, rapidly activating microglia and transforming them into an amoeboid, activated morphology [10,19,20]. Activated microglia initially have a protective role, as their migratory behavior helps clear apoptotic cell debris and limit the spread of damage [21,22]. However, persistent or excessive activation and migration can lead to their massive accumulation in the epileptogenic focus. They then release a large number of pro – inflammatory cytokines, such as IL – 1β and TNF – α [23], significantly lowering the seizure threshold, exacerbating blood – brain barrier disruption, synaptic dysfunction, and neuronal death, thus forming a vicious cycle of "seizure – neurodamage – neuroinflammation – re – seizure." [24,25].

The directed migration of microglia to the site of injury is a crucial step in the execution of their immune functions. This process is highly dependent on the specific expression of P2Y Purinoceptor 12 (P2RY12) on microglia [26,27]. P2RY12 is a G - protein - coupled receptor (GPCR) that mainly responds to extracellular "danger signals" such as ATP, regulating cell chemotaxis, phagocytosis, and the secretion of inflammatory factors [28] In cases of brain injury such as epilepsy, the extracellular ATP concentration rises sharply, acting as a key "find – me" signal recognized by P2RY12 [29]. Upon activation, P2RY12 activates Rho guanine nucleotide exchange factor (RhoGEF), which in turn activate RhoA and promote the activation of Rho – associated kinase (ROCK). Subsequently, ROCK regulates the reorganization of myosin light chains and the actin cytoskeleton, ultimately driving the directed

migration of microglia to the core of the injury site [30,31]. Therefore, targeting and inhibiting the function of P2RY12 is expected to precisely intervene in the pathological migration of microglia, thereby cutting off the above – mentioned neu-roinflammatory vicious cycle and providing a new strategy for epilepsy treatment.

Gastrodia elata is a traditional Chinese medicinal herb used to treat neurological diseases such as convulsions, dizzi-ness, and headaches. Its active ingredient, gastrodin, has been shown to have multiple neuroprotective effects [32–36]. A large number of preclinical studies have demonstrated that Gastrodin can reduce the intensity of epileptic seizures, delay the latency of seizures, and decrease the loss of hippocampal neurons in animal models [37–39]. In addition, pharma-cokinetic studies have shown that Gastrodin is rapidly absorbed orally, can cross the blood – brain barrier, and has good safety, indicating its high potential for drug development [40,41]. However, most current studies have focused on the terminal effects of Gastrodin on neuronal excitability or inflammatory factor expression. Its specific regulatory role in the upstream key event of microglia migration and its molecular target remain unclear. In particular, whether Gastrodin can directly act on the P2RY12 receptor, interfere with its mediated ATP chemotactic signaling and cytoskeletal remodeling, and thus inhibit the pathological recruitment of microglia still needs to be elucidated. Based on the above background, this study proposes the hypothesis: Gastrodin can directly target and bind to the P2RY12 receptor on microglia, inhibit the activation of its downstream RhoA/ROCK signaling pathway and the reorganization of F – actin cytoskeleton, thereby reducing the migration of microglia to the epileptogenic focus, and ultimately alleviating neuroinflammation and excessive neuronal damage in the lesion area. To verify this hypothesis, we integrated bioinformatics analysis, molecular docking simulations, and *in vitro* cell experiments to reveal the anti – epileptic mechanism of Gastrodin and provide a mechanistic basis for P2RY12 receptor as a new target for anti – epileptic treatment.

## 2. Materials and methods

### 2.1 Bioinformatics analysis

To systematically identify key genes related to the pathological progression of epilepsy, we obtained and analyzed the gene expression dataset GSE256068 from the NCBI – GEO database (https://www.ncbi.nlm.nih.gov/geo/). This dataset included 59 epilepsy samples with hippocampal sclerosis and 11 healthy control samples. We used R language (v4.2.0) and its limma package for data preprocessing. Through differentially expressed gene (DEG) analysis (adjusted P – value < 0.05, |log2FC| > 1), we screened out genes that were significantly up – regulated or down – regulated in the epi-lepsy group. We used the "ggplot2" R software package to draw a volcano plot to show their distribution and significance. To explore the biological functions of the differential genes, we used the "ClusterProfiler" package in R for Gene Ontology (GO) and Kyoto Encyclopedia of Genes and Genomes (KEGG) analyses, with a threshold P – value < 0.05. Through the GeneCards website (https://www.genecards.org/), we searched for epilepsy – related genes using "Pediatric epilepsy" as the keyword and obtained 2620 disease – related genes after screening with a relevance score > 4. Through BATMAN – TCM (http://bionet.ncpsb.org.cn/batman – tcm/index.php), we obtained the active ingredients and predicted targets of Gastrodia elata. We used a VENN diagram to show the intersection between DEGs, disease – related genes, and pre-dicted targets, and used the "Pheatmap" R software package to draw a heat map to show the expression distribution of these intersecting genes. To further explore the relationship between targets and active ingredients, we used Cytoscape to draw a Gastrodia elata active ingredient – drug target network. To optimize disease predictive markers, we used SVM – RFE, Boruta, and Lasso regression methods to screen key targets, providing a bioinformatics basis for subsequent exper-imental mechanism research. Taking the spearman correlation between P2RY12 and the expression levels of all other genes as input, we calculated the biological functions affected by P2RY12 based on GSEA. Using simplifyEnrichment, we calculated the correlation of the above functional enrichment results and performed clustering, and drew a functional network to show the functions associated with P2RY12. To identify the characteristics of immune cells in the brain tissues of normal samples and epilepsy patients, we used the single - sample gene set enrichment analysis (ssGSEA) algo-rithm to estimate the differences in the infiltration abundance of 28 immune cell types between the two groups based on

expression data [42]. Pearson correlation analysis revealed the correlation between the expression of key genes and the distribution of immune cells.

Molecular docking is a computational method used to predict the interactions and binding propensities between two or more molecules. It is commonly used in drug design and biomolecular research. By simulating the binding process of small molecules to macromolecules, it predicts the affinity of drug molecules to their target proteins. This method helps identify potential drug candidate molecules and understand the nature of molecular interactions. We downloaded the two – dimensional molecular structures of key compounds from the PubChem database (https://pubchem.ncbi.nlm.nih.gov/); downloaded the P2RY12 protein structure (4NTJ) from the Protein Data Bank (https://www.rcsb.org/). We used the computer – aided drug design software Schrodinger 2024 to obtain the optimal binding conformations of proteins and drugs. The software for visualizing molecular docking results was Pymol 2.3.0.

## 2.2  Cell culture and *in vitro* epilepsy model construction

The human microglia cell line HMC3 and the human neuroblastoma cell line SH - SY5Y were purchased from the Cell Resource Center of the Shanghai Institute of Life Sciences, Chinese Academy of Sciences. HMC3 cells were cultured in DMEM medium (Sigma – Aldrich, United States, D6429) containing 10% fetal bovine serum (Gibco, United States, A5669701) and 1% penicillin – streptomycin (Sigma – Aldrich, United States, P7539) under standard conditions of 37°C and 5% $CO_2$. SH - SY5Y cells were cultured in DMEM medium containing 10% FBS and 1% penicillin – streptomycin. Before inducing their differentiation, they were adapted for 2 days. When the cell confluence reached 60–70%, they were treated with 10 µM all – trans retinoic acid (RA) (Sigma – Aldrich, United States, R2625) for 5 days (the RA medium was replaced every 48 h). After removing the RA medium, a medium containing brain - derived neurotrophic factor (BDNF) (R&D Systems, USA, 11166 – BD) was added to induce differentiation for 3–5 days (the medium was half – replaced every 48 h) to induce them to differentiate into a neuron – like state.

To construct a co – culture model of human microglia and neurons to simulate the pathological situation of epilepsy *in vitro*, we first cultured microglia HMC3 in the upper chamber of a transwell, and neurons SH - SY5Y in the lower chamber. On this basis, we replaced the medium with a medium containing 100 µM kainic acid (KA) (Merck, Germany, K2389) and cultured for 24 h to construct an epilepsy model (EM) through chemical induction.

## 2.3  qRT - PCR experiment

Quantitative reverse transcription – polymerase chain reaction (qRT - PCR) technology was used to quantitatively detect the mRNA expression levels of target genes. Total RNA was extracted using TRIzol reagent (Thermo Fisher Scientific, USA, #15596026) according to the manufacturer's instructions. Then, the PrimeScript™ RT Reagent Kit (Takara Bio, Japan, RR047A) was used for reverse transcription to generate cDNA from RNA. Using cDNA as a template, qPCR amplification was performed using TB GreenPremix (Takara Bio, Japan, RR820A). The program was set as follows: pre – denaturation at 95°C for 30 seconds, followed by 30 cycles of denaturation at 95°C for 30 seconds and annealing/extension at 60°C for 30 seconds. The relative expression levels of target genes were calculated using the $2^{-\Delta\Delta Ct}$ method. All experiments were performed with three biological replicates to ensure data reliability. The primer sequences used in this study are presented in Supplementary Table 1.

## 2.4  Western blot experiment

Cells were collected, and RIPA lysis buffer (Beyotime, China, P0013B) containing protease inhibitors (Roche, Switzerland, 04693132001) and phosphatase inhibitors (Thermo Fisher Scientific, USA, 78420) was added. The cells were lysed on ice for 30 minutes. The samples were then centrifuged at 12,000 rpm at 4 °C for 15 minutes, and the supernatant was collected to obtain total protein. The protein concentration was determined using a BCA protein quantification kit (Thermo Fisher Scientific, USA, 23225). After adjusting the protein concentration according to the measurement results, 5 × SDS

loading buffer (Beyotime, China, P0015L) was added, and the samples were boiled at 95 °C for 5 minutes to denature the proteins. Subsequently, SDS-PAGE gels of appropriate concentrations were prepared for electrophoresis to separate the proteins. The proteins were then transferred to a PVDF membrane (Millipore, USA, IPVH00010), which was blocked with 5% skim milk (BD, USA, 232100) in TBST buffer (20 mM Tris-HCl, 150 mM NaCl, 0.1% Tween 20, pH 7.6) for 1–2 hours. Primary antibodies were then added: P2RY12 antibody (dilution ratio 1:1000), RhoA antibody (Cell Signaling Technology, USA, 2117, dilution ratio 1:1000), ROCK antibody (Cell Signaling Technology, USA, 4035, dilution ratio 1:1000), and GAPDH (Cell Signaling Technology, USA, 5174, dilution ratio 1:1000). The membrane was incubated with the primary antibodies overnight at 4 °C. The next day, the membrane was washed three times with TBST for 5 minutes each time. Secondary antibodies, goat anti-rabbit IgG (Beyotime, China, A0208, dilution ratio 1:1000) and goat anti-mouse IgG (Beyotime, China, A0216, dilution ratio 1:1000), were then added, and the membrane was incubated at room temperature for 1–2 hours. After another round of washing, the membrane was developed using an ECL chemiluminescent substrate (Thermo Fisher Scientific, USA, 34580) and imaged on a chemiluminescence imaging system (Bio-Rad, USA, ChemiDoc MP). The gray values of the bands were analyzed using ImageJ software. GAPDH was used as an internal reference to calculate the relative expression levels of the target proteins, and the expression differences among different treatment groups were compared.

## 2.5 Pull-down experiment

This experiment employed a pull-down method based on the biotin-streptavidin system to verify the direct binding of Gastrodin to the P2RY12 protein using whole-protein lysates from HMC3 cells and purified P2RY12 protein. First, biotin-labeled Gastrodin was incubated with streptavidin magnetic beads (Beyotime, China, P2151) at 4 °C for 2 hours for immobilization. Meanwhile, a free biotin group and a blank magnetic bead group were set up as negative controls. After washing the immobilized magnetic beads with PBST, they were incubated with the total protein lysate from HMC3 cells (prepared using RIPA lysis buffer (Beyotime, China, P0013B) containing protease inhibitors) and purified P2RY12 protein (CUSABIO, China, CSB CF861997HU) at 4 °C with rotation overnight. The next day, after strict washing with PBST, 1×SDS loading buffer was added, and the samples were boiled in a water bath for elution. The supernatant was collected for SDS-PAGE electrophoresis. Finally, Western blotting was performed using an anti-P2RY12 antibody (Thermo Fisher, USA, RAB02600) and an HRP-labeled goat anti-rabbit secondary antibody (Beyotime, China, A0208) to verify the specific pull-down of the P2RY12 protein.

## 2.6 ELISA experiment

ELISA kits (R&D Systems, USA, DTA00D for TNF-α detection; R&D Systems, USA, DLB50 for IL-1β detection) were used to detect the levels of TNF-α and IL-1β. All operations were carried out strictly according to the kit instructions. The 96-well ELISA plates were coated with the corresponding capture antibodies and incubated at 4 °C overnight. The next day, the coating solution was discarded, and the plates were washed three times with the washing buffer provided in the kit for 3 minutes each time. The blocking solution (provided in the kit) was added, and the plates were incubated at room temperature for 1 hour. After discarding the blocking solution and washing three times again, the diluted samples, including cell culture supernatants or lung tissue supernatants after homogenization (three biological replicates per group), were added, and the plates were incubated at room temperature for 2 hours. After washing three times, the biotin-labeled detection antibodies were added, and the plates were incubated at room temperature for 1 hour. After another round of washing, avidin-HRP was added, and the plates were incubated at room temperature for 30 minutes. After washing three times, the TMB substrate solution was added, and the plates were incubated in the dark for 15–30 minutes for color development. Finally, the stop solution was added to terminate the reaction. The absorbance was read at 450 nm using a microplate reader (BioTek, Synergy H1). Standard curves were prepared according to the kit instructions using the standard products provided in the kit, and the concentrations of TNF-α and IL-1β were calculated based on the standard curves.

## 2.7 Immunofluorescence

First, cells were collected, and a single-cell suspension was prepared using 0.25% trypsin-EDTA (Gibco, USA, #25200072). The cell density was adjusted to $1 \times 10^5$ cells/mL, and the cells were seeded on poly-L-lysine-coated slides (Thermo Fisher Scientific, USA, 15040−010) and cultured for 24 hours to form cell monolayers. After treatment, the cell monolayers were fixed with 4% paraformaldehyde for 15 minutes, washed with PBS, and then treated with 0.1% Triton X-100 for 10 minutes to increase cell permeability. Subsequently, the cells were blocked with 5% bovine serum albumin (BSA, Sigma, USA, A9647) for 1 hour. For F-actin staining, Alexa Fluor 568-labeled phalloidin (Phalloidin, Beyotime, C2201S) was directly added, and the cells were incubated in the dark at room temperature for 30 minutes. For the staining of MAP2, SYP, and P2RY12 proteins, primary antibodies were added: MAP2 (Abcam, USA, ab11267), SYP (Abcam, USA, ab32127), and P2RY12 (Thermo Fisher, USA, 702516), and the cells were incubated at 4 °C overnight. The next day, after washing with PBS, the corresponding fluorescently labeled secondary antibodies (Alexa Fluor 488-labeled secondary antibody and Alexa Fluor 594-labeled secondary antibody) were added, and the cells were incubated in the dark at room temperature for 1 hour. Then, DAPI staining solution (Beyotime, China, C1005, 1 µg/mL) was added, and the cells were incubated for 5 minutes to stain the nuclei. After washing with PBS, the images were acquired using a confocal microscope (Zeiss LSM880).

## 2.8 Calcium imaging experiment

The calcium imaging experiment was performed using a Fluo-4 AM fluorescent probe kit (Invitrogen, USA, F14201). First, SH-SY5Y and HMC3 cells were co-cultured in laser confocal culture dishes. After gently washing the cells with PBS, a working solution containing 5 µM Fluo-4 AM was added, and the cells were incubated in a 37 °C, 5% $CO_2$ incubator in the dark for 45 minutes. Then, the probe solution was removed, and the cells were incubated with fresh serum-free medium in the dark for an additional 20 minutes to ensure complete de-esterification of the probe. Real-time image acquisition was performed using a laser confocal microscope (Zeiss LSM 880) at an excitation wavelength of 488 nm. The time series was set to five time points: 0, 5, 10, 30, and 60 minutes. Three biological replicates were set up for each group. The image data were analyzed using ImageJ software to calculate the changes in fluorescence intensity, and the calcium transient frequency and peak intensity were calculated to assess neuronal excitability and calcium homeostasis changes.

## 2.9 Live/Dead cell staining experiment

Cell viability was detected using a Calcein-AM/PI cell viability/cytotoxicity detection kit (Beyotime, China, C2015M). After different treatments, the co-cultured HMC3 and SH-SY5Y cells were washed gently with PBS once. Then, a staining working solution containing 2 µM Calcein-AM and 1.5 µM PI was added, and the cells were incubated at 37 °C in the dark for 30 minutes. After incubation, the images were acquired using a fluorescence microscope (Nikon Eclipse Ti2): Calcein-AM showed green fluorescence (live cells) under excitation at 488 nm, and PI showed red fluorescence (dead cells) under excitation at 561 nm. The fluorescence intensity was quantitatively analyzed using ImageJ software to calculate the ratio of live cells to dead cells.

## 2.10 Protein thermal shift assay

The cellular thermal shift assay (CETSA) was used to verify the binding of Gastrodin to the P2RY12 protein. HMC3 cells were seeded in 6-well plates and reached 80% confluence. Then, they were divided into a non-drug group and a Gastrodin (100 µg/mL) treatment group and treated for 24 hours. After trypsinization, the cells were resuspended in PBS and aliquoted into 200 µL PCR tubes, with three replicates per group. The samples were heated at different temperature gradients (37 °C, 45 °C, 52 °C, 58 °C, 65 °C) for 3 minutes and immediately cooled on ice for 5 minutes. Then, three freeze-thaw cycles (quick freezing in liquid nitrogen – thawing at room temperature) were performed, and the samples were centrifuged to collect the supernatant as the soluble protein samples. The protein concentration was quantified using

 

the BCA method (Thermo Fisher, USA, 23225), and an equal amount of protein was subjected to Western blotting. The membrane was incubated with a P2RY12 antibody (Abcam, USA, ab254178) overnight, and an HRP-labeled secondary antibody was used for development. The band intensities of the P2RY12 protein in the drug-added and non-drug-added groups at different temperatures were compared to assess the changes in its thermal stability.

## 2.11 CCK-8 cell viability assay

The CCK-8 assay was performed using a Cell Counting Kit-8 (Dojindo, Japan, CK04). HMC3 and SH-SY5Y cells were seeded separately in 96-well plates at a density of $5 \times 10^3$ cells per well and cultured for 24 hours to allow cell attachment. A Gastrodin concentration gradient was set up (0, 10, 50, 100, 200, 500 μM), with six replicates per group and a blank well as a background control. After drug addition, the cells were cultured for an additional 24 hours. Then, 10 μL of CCK-8 solution was added to each well, and the plates were gently mixed and incubated at 37 °C for 2 hours. The absorbance values (OD values) of each well were measured at 450 nm using a microplate reader (BioTek Synergy H1). The cell viability was calculated using the following formula:

Cell viability (%) = (OD value of the experimental group – OD value of the blank group) / (OD value of the control group – OD value of the blank group) × 100%

The final results were expressed as the mean ± standard deviation of three independent experiments.

## 2.12 Data statistical methods

To ensure the accuracy and reliability of the experimental results, all data were subjected to rigorous statistical analysis. First, all experiments were independently repeated at least three times, and each experimental group contained at least three technical replicates. All data were presented as mean ± standard deviation (SD) from at least three independent experiments. Data analysis was performed using GraphPad Prism 10.0 software. Normality and homogeneity of variance were assessed using Shapiro–Wilk test and Levene's test, respectively. For comparisons between two groups, an unpaired two-tailed Student's t-test was used; for comparisons among three or more groups, one-way ANOVA followed by Tukey's multiple comparisons test was performed. If data failed normality or equal variance tests, the nonparametric Kruskal–Wallis test with Dunn's multiple comparisons test was used. All statistical analysis results were presented with significance levels of $P < 0.05$ (*), $P < 0.01$ (**), $P < 0.001$ (***), and $P < 0.0001$ (****). P-value < 0.05 was considered statistically significant. All significant markings in the charts were directly labeled on the figures or detailed in the figure legends.

## 3. Results

### 3.1 Bioinformatics analysis of DEGs and functional enrichment in epilepsy

We first performed DEGs on 59 epilepsy samples with human hippocampal sclerosis and 11 control samples in the GSE256068 dataset, using |log2FC| ≥ 1 and adjusted p-value < 0.05 as the thresholds. A total of 274 differentially expressed genes were identified, including 196 genes upregulated and 78 genes downregulated in the epilepsy samples. These DEGs may be potential key regulatory genes in the pathological process of epilepsy (Fig 1a). To further explore the biological processes and functional changes mediated by the DEGs, we conducted GO and KEGG enrichment analyses. The GO analysis mainly showed enrichment in biological functions such as immune cell migration and chemotaxis and immune response, suggesting that the migration and chemotaxis of immune cells, especially microglia, may be a key link in the pathogenesis of epilepsy (Fig 1b). The KEGG pathway enrichment analysis revealed that the DEGs were enriched in immune regulation and inflammation-related pathways (Fig 1c), indicating that the occurrence and development of epilepsy may be closely related to the abnormal activation of the immune system. Moreover, the abnormal neuroimmune response may lead to the exacerbation of the epilepsy course through persistent inflammation, immune cell migration, and the release of pro-inflammatory factors.

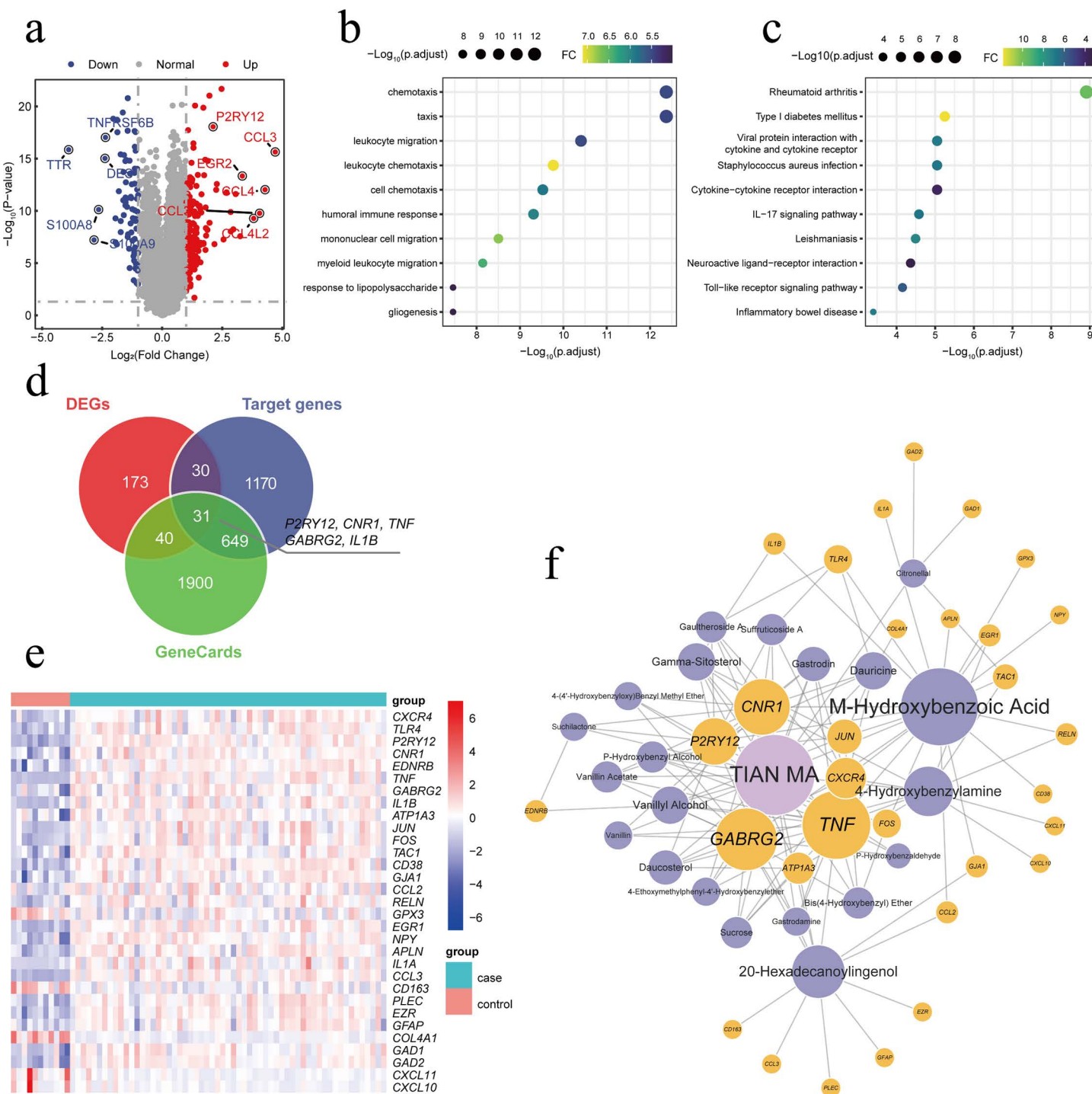

**Fig 1. Bioinformatics analysis of differentially expressed genes in epilepsy and the network of active ingredients and their targets of Gastrodia elata.** (a) The volcano plot shows the DEGs between epilepsy patients and healthy controls. Red represents significantly upregulated genes, and blue represents significantly downregulated genes. (b) GO enrichment analysis of DEGs. (c) KEGG enrichment analysis of DEGs. (d) Venn diagram illustrating the intersection of DEGs, epilepsy-related genes, and predicted targets of Gastrodia elata. (e) Heatmap showing the expression of the selected candidate targets between patients with epilepsy and healthy controls. (f) Network of active ingredients and drug targets of Gastrodia elata. Purple nodes represent active ingredients of Gastrodia elata, yellow nodes represent core genes in epilepsy, and lines represent interaction relationships.

To identify candidate drug targets of Gastrodia elata in epilepsy treatment, this study integrated multi-source data for analysis. First, 2,620 epilepsy-related disease genes were obtained from the GeneCards database, and the active components of Gastrodia elata and their predicted drug targets were screened using the BATMAN-TCM database (Fig 1d). Subsequently, the intersection of differentially expressed genes, disease-related genes, and the predicted targets of Gastrodia elata was taken to obtain candidate targets for Gastrodia elata in the treatment of epilepsy (Fig 1d). Furthermore, the expression distribution of these candidate targets was presented (Fig 1e). Among them, a total of 31 core candidate targets were identified in Fig 1d, including the key target P2RY12. To further explore the relationship between the targets and active ingredients, we constructed a Gastrodia elata active ingredient-drug target network. Among them, *CNR1, P2RY12, TNF, and GABRG2* were targeted by most of the active ingredients. *CNR1* is involved in the regulation of neuronal excitability and the inhibition of neuroinflammation, *P2RY12* is associated with synaptic transmission and nerve injury repair, TNF is a core factor in the neuroinflammatory pathway, and GABRG2 directly regulates neurotransmitter balance. The functions of these targets are all closely related to the key pathological processes of epilepsy, suggesting that they may be core targets of Gastrodia elata in the intervention of epilepsy. Additionally, M-Hydroxybenzoic Acid targeted the most candidate targets, indicating that it may play a key role in the process of Gastrodia elata affecting epilepsy (Fig 1f).

### 3.2 P2RY12 is a potential therapeutic target for epilepsy

To further screen candidate targets, this study employed multiple machine learning methods to analyze 31 candidate genes. First, Lasso regression analysis ($\lambda$ min = 0.0009) initially identified 9 key genes (Fig 2a-2b). Subsequently, based on the SVM-RFE machine learning method, it was found that the model accuracy was highest when the number of features was 2, and increasing the number of features reduced the accuracy, indicating that streamlining the targets is beneficial for optimizing model performance (Fig 2c). Additionally, the Boruta algorithm in random forests further confirmed that 24 candidate genes had significant feature importance (Fig 2d). Combining the results of the three feature selection models, *P2RY12* was identified as a key target in this study (Fig 2e).

To investigate the potential functions of P2RY12 in epilepsy, this study calculated the expression correlation between P2RY12 and all other genes. *P2RY12* showed a significant positive correlation with genes such as *CX3CR1* and *TLR7* (Fig 3a). These genes are involved in the regulation of neuroinflammation and microglia activation, and neuroinflammation is an important pathological mechanism underlying seizures and brain damage, suggesting that P2RY12 may influence the progression of epilepsy by regulating neuroinflammatory responses and microglia activation. Subsequently, we further calculated the biological functions affected by P2RY12 based on GSEA and used simplifyEnrichment to perform correlation analysis and clustering of the functional enrichment results. The functional network showed that the functional modules associated with P2RY12 included synaptic pruning, macrophage activation, and were also involved in biological processes such as gliogenesis, lysosomal transport, and autophagosome maturation, suggesting that P2RY12 may participate in the pathological process of epilepsy through multidimensional coordinated regulation of neurodevelopment, cellular homeostasis, and immune inflammation (Fig 3b). Given that persistent neuroinflammation is an important pathological link in promoting seizures and exacerbating brain damage, we further analyzed the expression correlation between P2RY12 and multiple pro-inflammatory key genes, including *TREM2, CX3CR1, ITGAM, SPI1*, and *IRF8*, and found that P2RY12 showed a significant positive correlation with these pro-inflammatory genes. This result further supports the close correlation between *P2RY12* and neuroinflammatory status, suggesting that it may play an important role in the neuroinflammatory pathological process associated with epilepsy (Fig 3c).

Neuroinflammation in epilepsy is not limited to the internal environment of the CNS. Persistent inflammatory status can lead to the disruption of the blood-brain barrier, thereby causing the infiltration of peripheral immune cells, which, together with resident microglia, constitute the unique immune microenvironment of the epileptic focus. To further clarify the regulatory effect of P2RY12 on the immune microenvironment in epilepsy, we calculated the distribution of immune cells between the epilepsy sample group and the healthy control group and evaluated the correlation between P2RY12

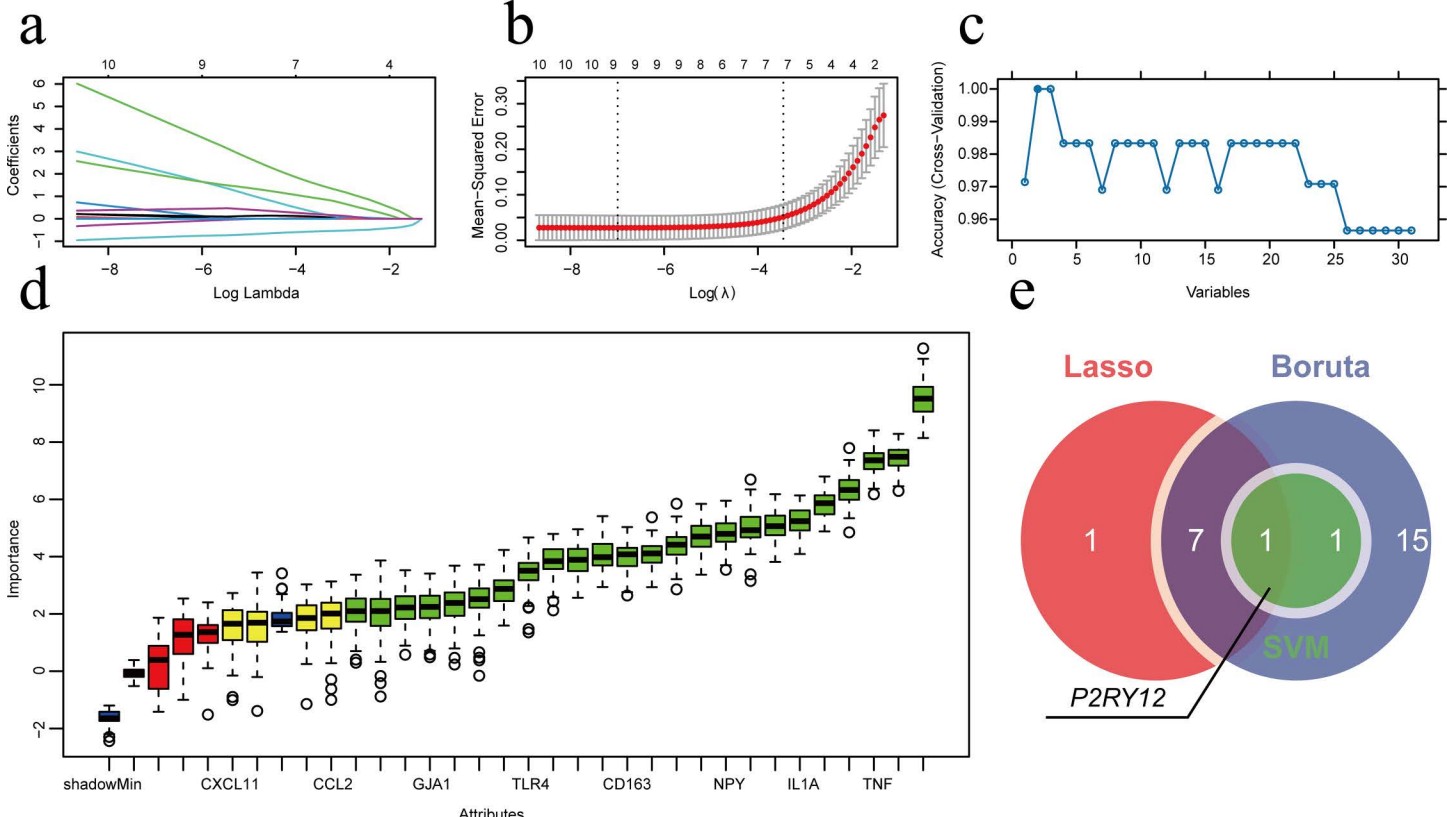

**Fig 2. Candidate target screening based on multiple machine learning methods.** (a) Lasso regression coefficient path plot showing the shrinkage path of the regression coefficients of 31 candidate genes as the parameter λ changes. (b) Lasso regression cross-validation error curve. The dashed lines represent λ.min and λ.1se, respectively, and 9 feature genes were finally selected. (c) Feature screening accuracy curve of the SVM-RFE algorithm. The y-axis represents the accuracy of ten-fold cross-validation, and the x-axis represents the number of features. The model accuracy was highest when the number of features was 2. (d) Feature importance distribution plot of the Boruta algorithm. It shows the Z-value distribution of all features after iterative calculations. The green box represents "confirmed features" (24), the red box represents "rejected features," and the yellow box represents "tentative features." (e) Venn diagram of the feature screening results of the three machine learning algorithms. P2RY12 is the key target selected by all three methods.

expression levels and the distribution of different immune cells. The heatmap results showed that the epilepsy sample group exhibited enhanced infiltration of effector memory CD4+ T cells, activated B cells, and activated CD4+ T cells, suggesting that the immune cell landscape undergoes significant remodeling in the epileptic state. The enhanced infiltration of these immune cells may be closely related to the imbalance of the immune microenvironment and the activation of inflammatory responses in the epileptic state (Fig 3d). Further correlation analysis revealed that the expression level of the *P2RY12* gene showed a significant positive correlation with the infiltration proportions of effector memory CD4+ T cells, activated B cells, and activated CD4+ T cells, suggesting that the high expression of *P2RY12* may participate in the regulation of the immune microenvironment in epilepsy by synergistically recruiting and activating peripheral immune cells, thereby playing an important role in the pathogenesis of epilepsy (Fig 3e).

### 3.3 Specific binding of the active ingredient of gastrodia elata, gastrodin, to the P2RY12 receptor protein

Based on the previously constructed interaction network of "active ingredients of Gastrodia elata – drug targets" (Fig 1f), we identified multiple active ingredients of Gastrodia elata with potential binding ability to the key target P2RY12 (including

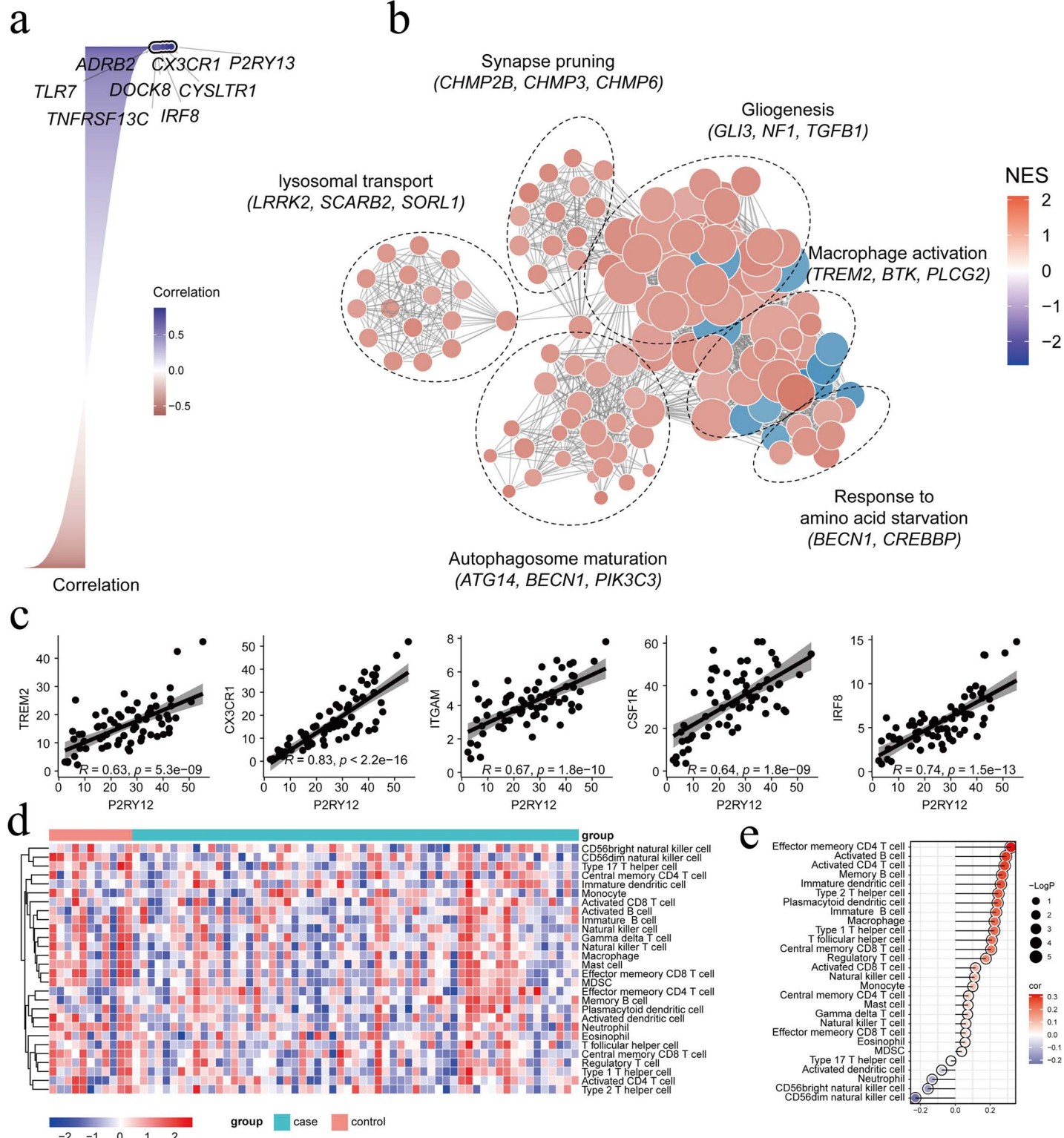

**Fig 3. Gene expression correlation and functional clustering analysis related to *P2RY12*.** (a) Expression correlation analysis between *P2RY12* and 30 other genes. Blue represents positive correlation, red represents negative correlation, and the darker the color, the stronger the correlation. (b)

Functional clustering analysis of *P2RY12*. Node colors represent the normalized enrichment score (NES), with red being positive and blue being negative. The darker the color, the larger the absolute value of the NES. (c) Expression correlation analysis between *P2RY12* and pro-inflammatory genes. (d) Heatmap of immune cell distribution in patients with epilepsy and healthy controls. (e) Correlation between P2RY12 expression levels and immune cell distribution.

Suffruticoside A, Suchilactone, Gamma-Sitosterol, M-Hydroxybenzoic Acid, 20-Hexadecanoylingenol, Gaultheroside A, Dauricine, Gastrodin, Vanillin Acetate, Daucosterol). To thoroughly evaluate their binding characteristics, this study systematically analyzed the interaction modes and binding free energies between these active ingredients and the P2RY12 protein using molecular docking technology. The results showed that most of the tested ingredients could bind to the active pocket of P2RY12 in a stable conformation, with binding free energies ranging from −2.7 to −8.3 kcal/mol. Among them, Gastrodin exhibited the lowest binding free energy (−8.3 kcal/mol), indicating that it had the most stable binding conformation with P2RY12 and could bind to the active pocket of the target in the most energetically favorable conformation. This result suggests that Gastrodin is the most potential candidate active ingredient for binding to P2RY12, providing a basis for subsequent research on its functional regulatory effects (Fig 4a-4k).

To further confirm the specific interaction between Gastrodin and the P2RY12 protein *in vitro*, we used the Pull-down technique to verify this binding. First, we performed a Pull-down experiment using the whole-cell lysate of HMC3. The results showed that endogenous P2RY12 protein was detected in the pulled-down fraction after treatment with biotin-labeled Gastrodin (Fig 5a). To exclude potential interference from other proteins in the cells and verify the directness of this interaction, we further conducted the experiment using purified recombinant P2RY12 protein. The results showed that Gastrodin could directly bind to the recombinant P2RY12 protein, while no obvious bands were observed in the control group (Fig 5b). To verify this binding in a cellular environment closer to the physiological state, we used the cellular thermal shift assay (CETSA). The results showed that the thermal stability of the P2RY12 protein in HMC3 cells treated with Gastrodin was significantly higher than that in the control group, indicating that Gastrodin binding could enhance the conformational stability of P2RY12 (Fig 5c-5d). The above multi-level experimental results collectively confirmed that Gastrodin could directly and specifically bind to the P2RY12 protein *in vitro*.

### 3.4 Gastrodin alleviates neuronal calcium overload in the epilepsy model

A core pathological feature of seizures is the abnormal excitability of neurons and calcium homeostasis imbalance, which is considered a key mechanism underlying seizure-related neuronal damage and death [43]. To investigate whether Gastrodin can alleviate calcium overload in the epilepsy model, we first used the CCK-8 assay to evaluate the safe concentration range of Gastrodin for HMC3 microglia and SH-SY5Y neurons. The results showed that within the concentration range of 0–100 μM, Gastrodin did not produce significant toxicity to either cell type, and the cell viability remained above 95%. However, when the concentration of Gastrodin gradually increased to 500 μM, the viability of both HMC3 and SH-SY5Y cells significantly decreased (Fig 6a). Based on these experimental results, 100 μM was selected as the intervention concentration of Gastrodin for subsequent experiments.

After determining the safe intervention concentration of Gastrodin, we constructed an *in vitro* epilepsy model to simulate the neuroinflammatory environment *in vivo*. To obtain a cell model closer to mature neurons, we used RA and BDNF to induce SH-SY5Y cells to differentiate into cells with a neuronal-like phenotype. Subsequently, the differentiated SH-SY5Y cells and HMC3 cells were co-cultured in a Transwell co-culture system to reproduce the close communication relationship between neurons and microglia in the brain. By adding KA stimulation to this co-culture system, an *in vitro* epilepsy model (Epilepsy Model, EM) was established (Fig 6b). Then, we used calcium imaging technology to detect the dynamic changes of calcium ions in neurons in the co-culture system. The results showed that compared with the CTRL group, the EM group showed a significant increase in calcium influx 10 minutes after KA stimulation ($p < 0.0001$), indicating that the neurons were in a highly excited state, and the model construction was successful. After Gastrodin intervention

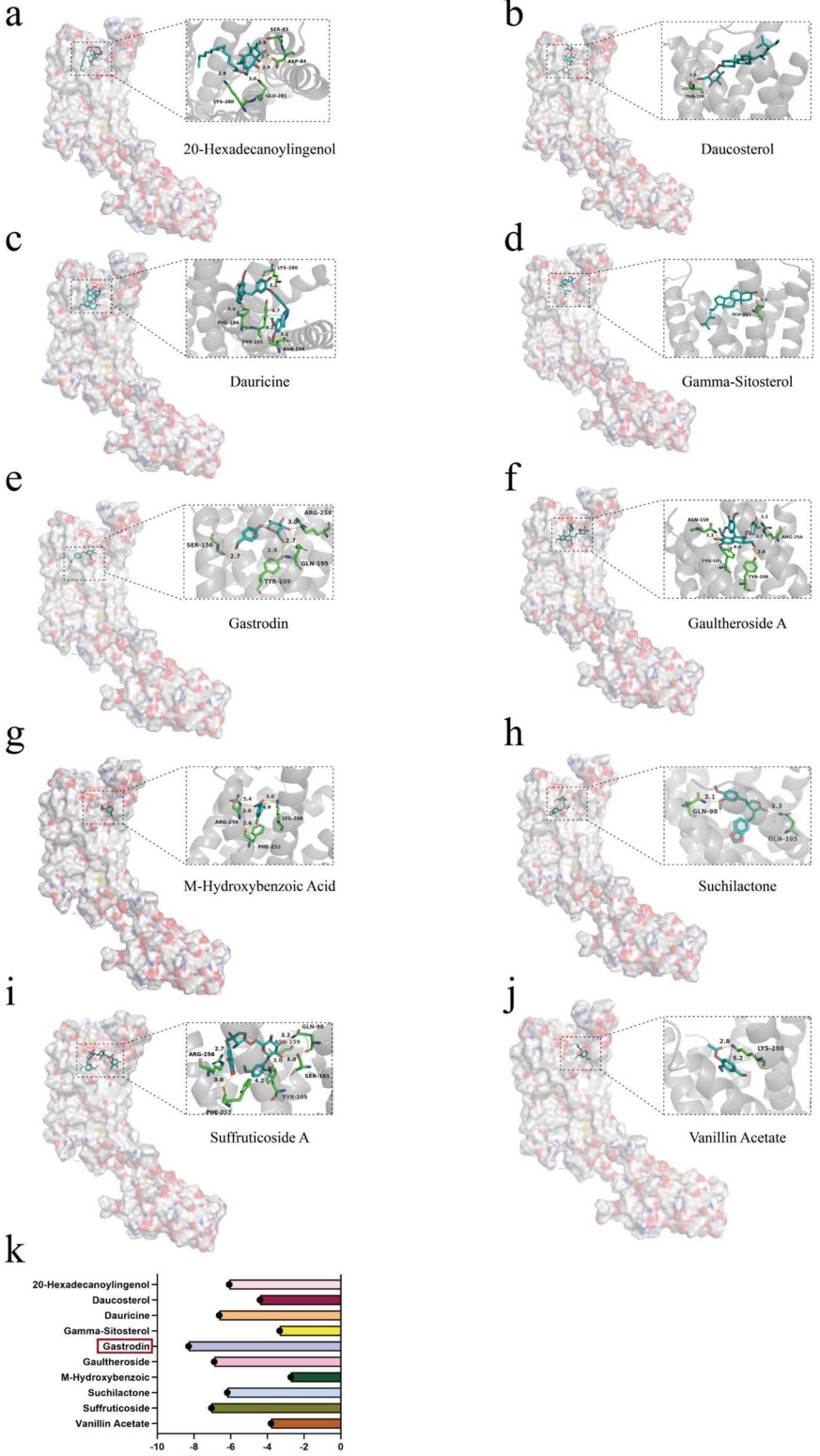

**Fig 4. Molecular docking results of active ingredients of Gastrodia elata with the P2RY12 protein.** (a-j) Molecular docking models of active ingredients of Gastrodia elata (Suffruticoside A, Suchilactone, Gamma-Sitosterol, M-Hydroxybenzoic Acid, 20-Hexadecanoylingenol, Gaultheroside A,

Dauricine, Gastrodin, Vanillin Acetate, Daucosterol) with the P2RY12 protein, showing the binding conformations of the active ingredients in the active pocket of the P2RY12 protein and key interactions. (k) Binding free energies of various active ingredients of Gastrodia elata with the P2RY12 protein.

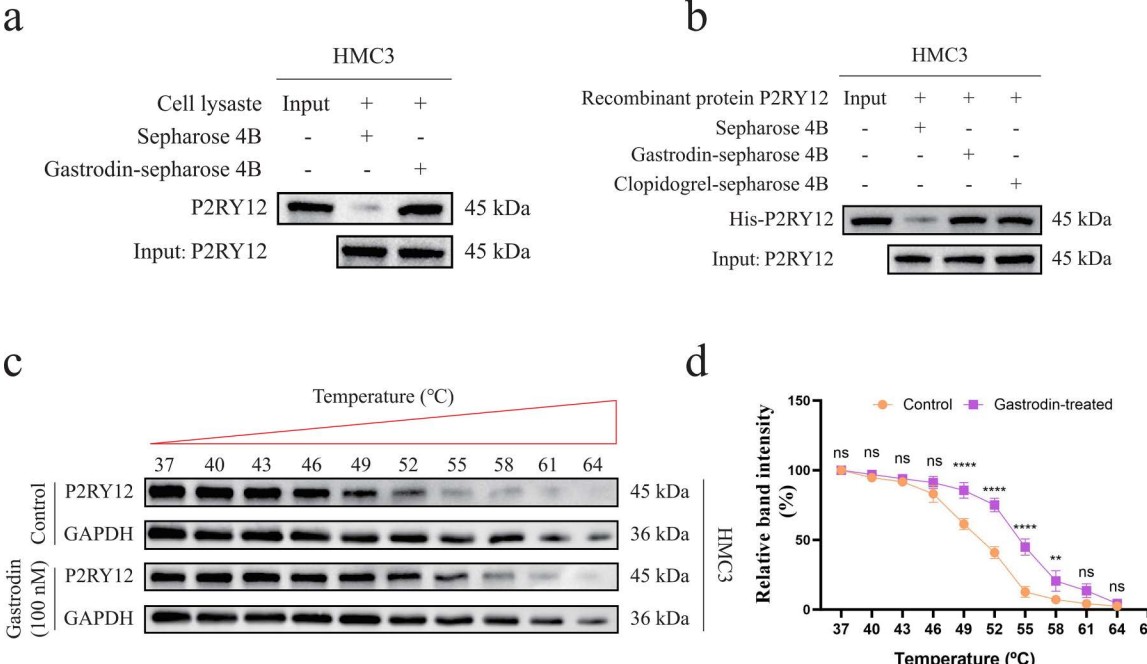

**Fig 5. Verification of the binding between Gastrodin and the P2RY12 protein.** (a-b) Pull-down experiments using the whole-cell lysate of HMC3 and recombinant P2RY12 protein to verify the specific binding between Gastrodin and the P2RY12 protein. (c-d) CETSA to verify the binding efficiency between Gastrodin and the P2RY12 protein.

(EM+Gastrodin group), the amplitude of calcium influx induced by KA stimulation at 10 minutes was significantly lower than that in the EM group (p < 0.001) (Fig 6c-6d). This result indicates that Gastrodin can effectively alleviate the calcium homeostasis disorder caused by epileptiform discharge, thereby inhibiting neuronal hyper-excitability.

### 3.5 Gastrodin reduces neuronal structural damage and cell death in the epilepsy model

During the process of seizures, in addition to abnormal electrophysiological activities [44], neurons often suffer from severe structural damage and apoptosis [45]. To further evaluate the neuroprotective effect of Gastrodin, we detected the structural integrity and survival of neurons. Immunofluorescence results showed that compared with the CTRL group, the expression of the synaptic protein Synaptophysin (SYP) in the EM group was significantly reduced (p < 0.0001), indicating that the epilepsy model caused significant synaptic damage. At the same time, the expression of the neuronal dendritic marker Microtubule-Associated Protein 2 (MAP2) also significantly decreased (p < 0.01), suggesting that the neuronal dendritic network was simultaneously damaged. After Gastrodin treatment (EM+Gastrodin group), the expression of MAP2 and SYP was significantly restored compared with the EM group, indicating that Gastrodin has a protective effect on neuronal structure and synaptic function (Fig 6e-6g). To further verify the effect of Gastrodin on cell survival, we used the Calcein-AM/PI double staining method for evaluation. The results showed that compared with the CTRL group, the proportion of live cells labeled with Calcein-AM in the EM group significantly decreased, while the proportion of dead cells

 

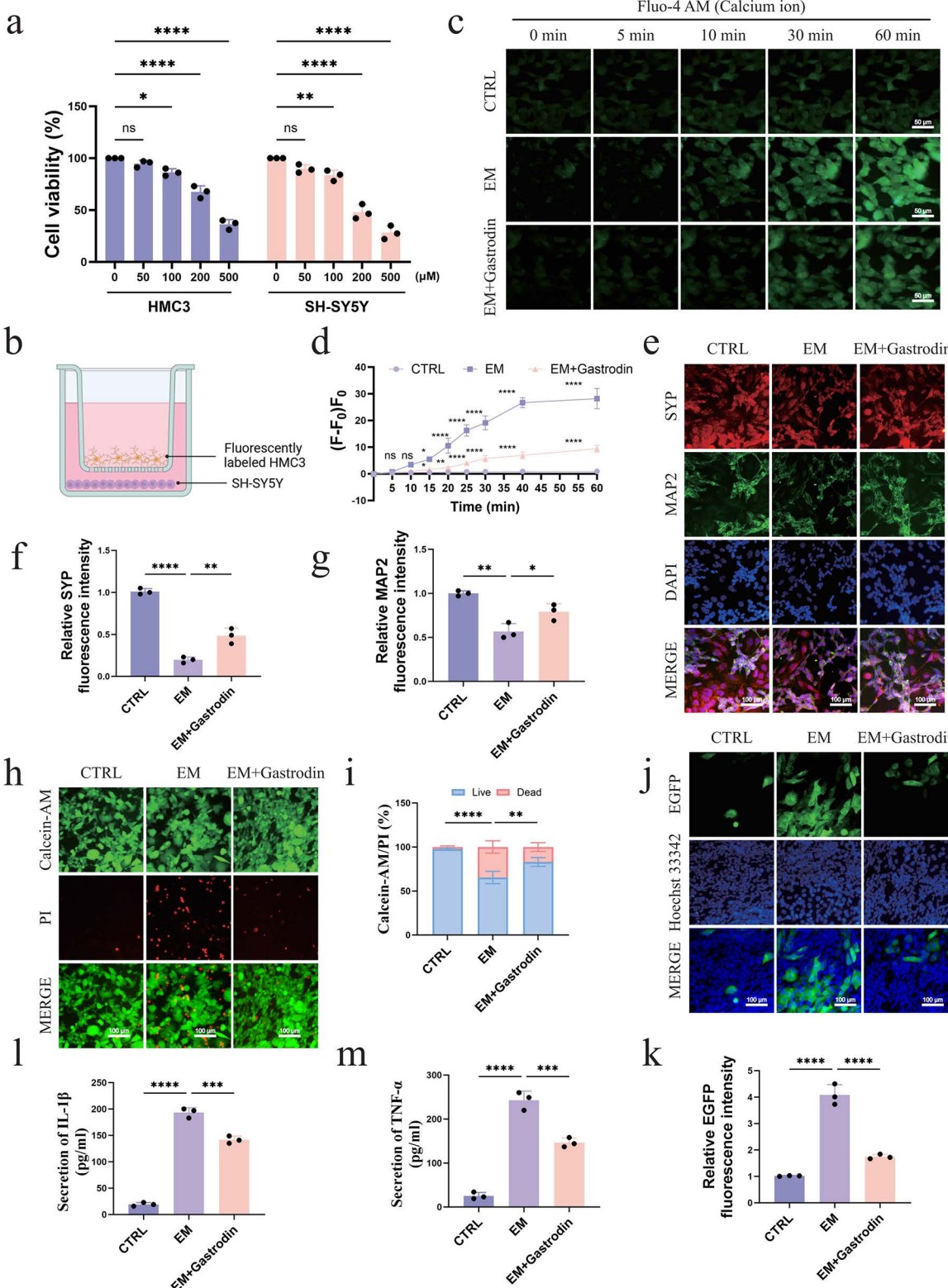

**Fig 6. Gastrodin alleviates seizure-related neuronal damage by inhibiting calcium overload and neuroinflammation.** (a) Toxicity of Gastrodin to HMC3 and SH-SY5Y cells detected by the CCK-8 assay (n = 6). (b) Schematic diagram of epilepsy model construction. (c-d) Calcium imaging to detect

the dynamic changes of calcium ions in neurons in the co-culture system, n = 3, scale bar = 50 μm. (e-g) Immunofluorescence to detect the expression of neuronal structural proteins (MAP2, SYP), n = 3, scale bar = 100 μm. (h-i) Calcein-AM/PI double staining to evaluate cell survival, n = 3, scale bar = 100 μm. (j-k) Immunofluorescence to detect the migration behavior of microglia, n = 3, scale bar = 100 μm. (l-m) ELISA to detect the secretion levels of the pro-inflammatory factors TNF-α and IL-1β in the culture supernatant, n = 3.

labeled with PI significantly increased (p < 0.0001), indicating that epilepsy can induce an increase in neuronal death. The Gastrodin treatment group significantly increased the proportion of live cells and decreased the proportion of dead cells (p < 0.01), indicating that Gastrodin can effectively reduce neuronal death in the epilepsy model and further confirming its neuroprotective effect on neurons (Fig 6h-6i).

### 3.6 Gastrodin inhibits microglia migration and reduces inflammatory responses

The abnormal migration and focal accumulation of microglia after seizures are key links in the occurrence and exacerbation of neuroinflammation [20]. The chemotactic signaling mediated by P2RY12 drives microglia to rapidly accumulate in the neuronal damage area. During this process, they transform from a resting state to an activated state and secrete a variety of pro-inflammatory factors, thereby amplifying the local inflammatory response [46]. Therefore, we first detected the effect of Gastrodin on microglia migration and further evaluated its regulatory effect on the secretion of inflammatory factors. In the *in vitro* epilepsy model, we used EGFP-labeled HMC3 microglia and observed their migration behavior through immunofluorescence. The results showed that compared with the CTRL group, microglia in the EM group significantly accumulated in the neuronal area (p < 0.0001), while after Gastrodin intervention, the migration of microglia in the EM+Gastrodin group was significantly reduced compared with the EM group (p < 0.0001) (Fig 6j-6k). This result indicates that Gastrodin can effectively inhibit the abnormal chemotaxis of microglia in the epileptic state and prevent their excessive accumulation in the focal area.

Subsequently, we detected the secretion levels of the typical pro-inflammatory factors TNF-α and IL-1β in the epilepsy model. ELISA results showed that compared with the CTRL group, the levels of the inflammatory factors TNF-α and IL-1β in the EM group were significantly increased (p < 0.001). After Gastrodin intervention, the levels of inflammatory factors in the EM+Gastrodin group were significantly downregulated compared with the EM group (p < 0.001) (Fig 6l-6m). Combined with the results of the migration experiment, this suggests that Gastrodin reduces the focal accumulation of microglia and inhibits the secretion of inflammatory factors after their activation, thereby alleviating the neuroinflammatory response associated with epilepsy.

### 3.7 Gastrodin inhibits microglia migration by regulating the downstream signaling pathway of P2RY12

To thoroughly investigate the molecular mechanism by which Gastrodin inhibits microglia migration, we focused on the downstream signaling pathway mediated by P2RY12. We detected the expression levels of P2RY12 and its downstream key signaling molecules. Western blot results showed that compared with the CTRL group, the expression of P2RY12 in the EM group was significantly upregulated (p < 0.0001), and the expression of its downstream signaling molecules RhoA and ROCK was also significantly enhanced (p < 0.0001), indicating that the P2RY12 pathway was abnormally activated. After Gastrodin treatment, although the total protein expression level of P2RY12 in the EM+Gastrodin group did not change significantly, the expression levels of RhoA and ROCK were significantly inhibited (RhoA, p < 0.0001; ROCK, p < 0.001), indicating that Gastrodin mainly exerts its effect by blocking the downstream signal transduction of P2RY12 rather than regulating the protein expression level of P2RY12. This inhibitory effect is consistent with the trend of the effect of a P2RY12-specific antagonist (Clopidogrel), further supporting that Gastrodin can regulate its downstream signals by directly targeting the P2RY12 receptor (Fig 7a-7b).

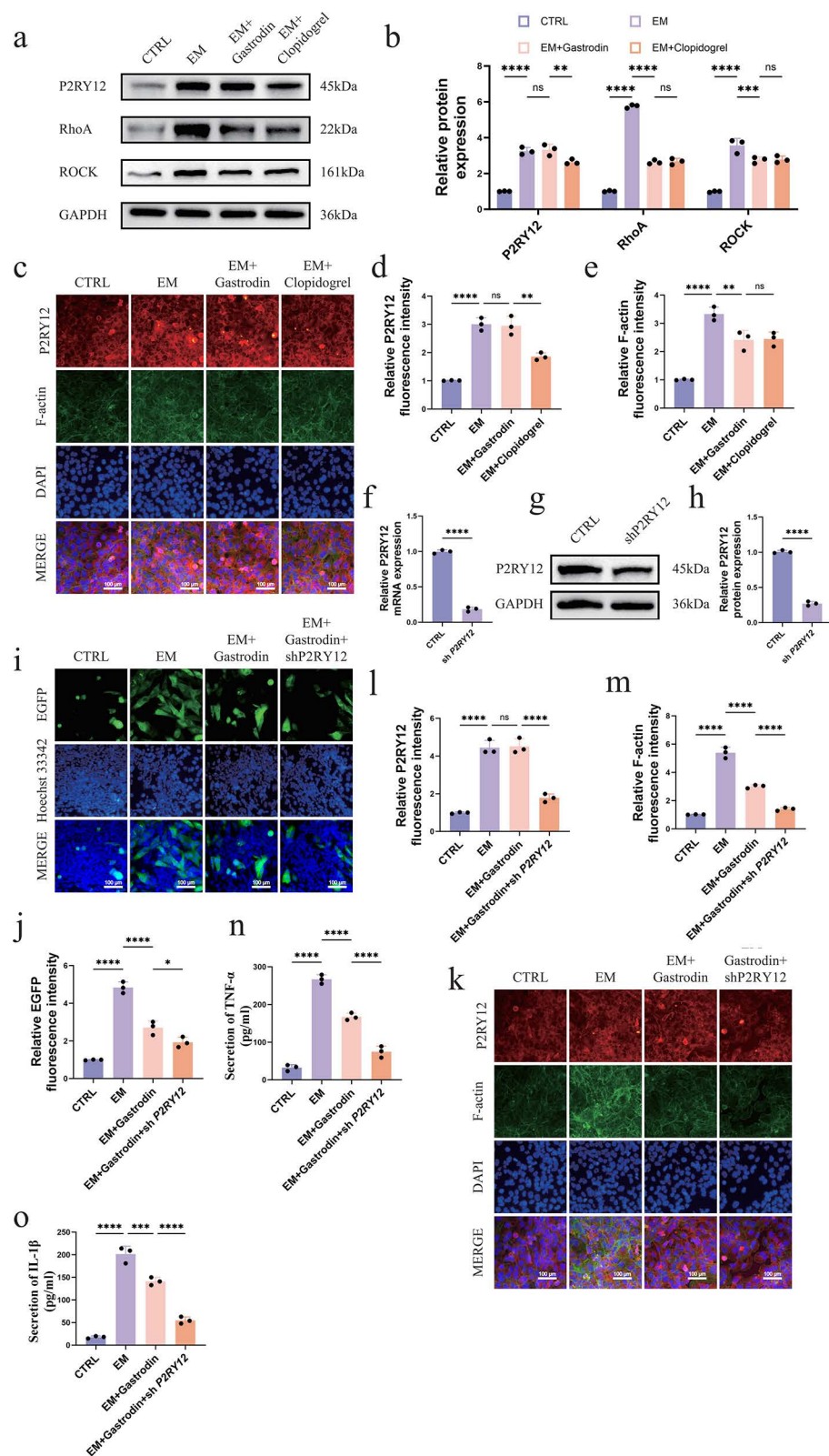

**Fig 7. Gastrodin inhibits microglia chemotactic migration and inflammatory responses by targeting the RhoA/ROCK signaling pathway.** (a-b) Western blot to detect the protein expression of P2RY12, RhoA, and ROCK, n = 3. (c-e) Immunofluorescence to detect the expression of P2RY12 and

F-actin, n = 3, scale bar = 100 µm. (f-g) Immunofluorescence to detect the migration behavior of microglia, n = 3, scale bar = 100 µm. (h-j) Immunofluorescence to detect the expression of P2RY12 and F-actin, n = 3, scale bar = 100 µm. (k-l) ELISA to detect the secretion levels of the pro-inflammatory factors TNF-α and IL-1β in the culture supernatant, n = 3.

To further investigate the regulation of this pathway on the microglia cytoskeleton, we detected the aggregation state of F-actin through immunofluorescence staining. The results showed that compared with the CTRL group, the fluorescence intensity of F-actin in the EM group was significantly higher than that in the control group ($p < 0.0001$), indicating that epilepsy induced significant cytoskeletal reorganization in microglia, thereby enhancing their migration ability. After Gastrodin treatment, the fluorescence intensity of F-actin in the EM+Gastrodin group was significantly lower than that in the EM group ($p < 0.01$), indicating that Gastrodin can effectively inhibit the cytoskeletal reorganization induced by epilepsy. Notably, the P2RY12-specific antagonist treatment group also showed a similar inhibition of F-actin aggregation as Gastrodin, further supporting that Gastrodin may inhibit microglia migration by regulating cytoskeletal dynamic changes (Fig 7c-7e).

### 3.8 P2RY12 knockdown inhibits microglia migration and inflammatory factor secretion

P2RY12 plays a key regulatory role in microglia migration and inflammatory responses. To further elucidate its function, we constructed a *P2RY12* knockdown HMC3 microglia cell line (sh *P2RY12*). Western blot and qPCR results showed that the mRNA and protein expression levels of P2RY12 in the sh *P2RY12* group were significantly reduced, indicating that the model construction was successful (S1 Fig). Subsequently, we evaluated the effect of *P2RY12* knockdown on the therapeutic effect of Gastrodin. Compared with the EM+Gastrodin group, the expression of EGFP in the EM+Gastrodin+sh *P2RY12* group was significantly downregulated ($p < 0.05$) (Fig 7f-7g), and the polymerization of F-actin was significantly inhibited ($p < 0.0001$) (Fig 7h-7j), accompanied by a significant decrease in the secretion of the inflammatory factors TNF-α and IL-1β ($p < 0.0001$) (Fig 7k-7l). The above results indicate that P2RY12 knockdown can further enhance the inhibitory effects of Gastrodin on microglia migration, cytoskeletal reorganization, and inflammatory responses, suggesting that the neuroprotective effect of Gastrodin is at least partially dependent on its targeting of the P2RY12 receptor. The weakening of P2RY12 function and the pharmacological intervention of Gastrodin show a synergistic effect in inhibiting neuroinflammation, further establishing P2RY12 as a key molecular target for Gastrodin to exert its anti-epileptic effect. In summary, our study shows that during seizures, the expression of the P2RY12 protein is significantly upregulated. After being activated by danger signals such as extracellular ATP, it initiates the downstream RhoA/ROCK signaling pathway, triggers the reorganization of the F-actin cytoskeleton, drives microglia to migrate to the focal area, and synergistically promotes the release of inflammatory factors such as TNF-α and IL-1β in this process, jointly exacerbating the neuroinflammatory response and ultimately leading to neuronal damage. Gastrodin can directly target and bind to the P2RY12 protein, inhibit the activation of its downstream signaling network, and block the pathological migration of microglia, thereby protecting neurons and exerting its anti-epileptic effect (Fig 8).

## 4. Discussion

Epilepsy is a complex neurological disease, and its pathological process is far from being simply abnormal neuronal discharge activity. Neuroinflammation has been proven to be an important factor driving the occurrence and development of epilepsy [47]. As resident immune cells in the CNS, microglia are rapidly activated after seizures. In the early stage, microglia mediate chemotactic migration to the focal area through the P2RY12 receptor sensing the ATP gradient released by neurons [29,48]. Subsequently, the aggregated microglia are further activated by inflammatory receptors such as TLR4 and NLRP3 under continuous pathological stimulation, transform into a pro-inflammatory phenotype, and release a large amount of pro-inflammatory factors such as TNF-α and IL-1β, exacerbating the excitotoxic damage and synaptic dysfunction of neurons [49–51]. Therefore, intervening in the pathological activation and migration of microglia, especially targeting its key receptor P2RY12 for regulation, has become a highly potential new strategy for anti-epileptic treatment.

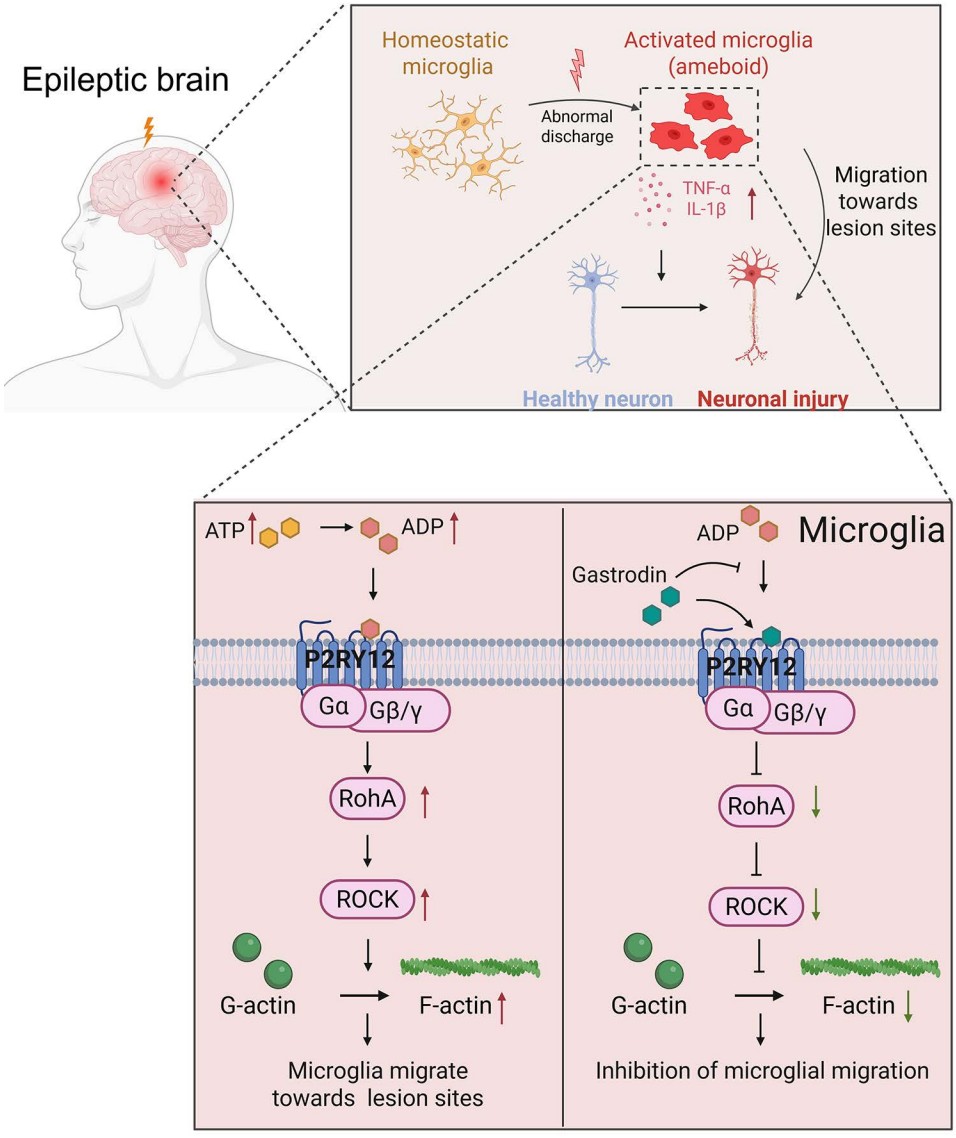

**Fig 8. Schematic diagram of gastrodin targeting P2RY12 to inhibit RhoA/ROCK pathway and microglial migration in epileptic models.**

Against this background, the anti-epileptic potential of traditional Chinese medicine Gastrodia elata and its active ingredients has attracted increasing attention. Gastrodin, the main bioactive ingredient of Gastrodia elata, has a wide range of neuroprotective effects, especially in the regulation of neuroinflammation [35, 52]. For example, in Alzheimer's disease research, Li et al. found that Gastrodin can significantly reduce the neuroinflammatory level in the brains of 3xTg-AD transgenic mice by inhibiting the excessive activation of the NF-κB/NLRP3 inflammatory signaling axis in microglia [34]. Zhang et al. also reported that Gastrodin can induce the polarization of microglia to the anti-inflammatory M2 phenotype through the Nrf2 pathway, thereby exerting a neuroprotective effect [35]. However, whether Gastrodin exerts its effect by regulating microglia function in epilepsy and its specific molecular mechanism remain to be elucidated.

Based on this, this study thoroughly investigated the pharmacological effects and molecular mechanisms of Gastrodin, the main active ingredient of the traditional Chinese medicine Gastrodia elata, in epilepsy and related neuroinflammation through a multi-level experimental system. We integrated bioinformatics prediction, molecular docking simulation, and functional experimental verification to systematically reveal that Gastrodin directly targets the purinergic receptor P2RY12 on microglia and effectively inhibits the activation of its downstream RhoA signaling pathway. This finding not only clarifies the direct target of Gastrodin at the receptor level but also provides new insights into its regulation of neuroimmune responses.

Further mechanistic studies showed that Gastrodin significantly inhibits the migration and accumulation of microglia to the neuronal damage area by regulating the reorganization of the F-actin cytoskeleton. At the same time, ELISA data showed that Gastrodin treatment significantly reduces the secretion of the pro-inflammatory factors TNF-α and IL-1β, suggesting that it has multiple effects in regulating immune balance. In terms of neurons, calcium imaging results showed that Gastrodin can significantly alleviate the abnormal discharge activity and structural damage of neurons, demonstrating a clear neuroprotective effect. Notably, through the P2RY12 knockdown experiment mediated by shRNA, this study proved the necessity of this receptor in the pharmacological effect of Gastrodin from the perspective of loss of function and provided experimental support for the precise targeting treatment potential of Gastrodin.

Additionally, despite the fact that this study, through bioinformatics analysis, molecular docking, pull-down assays, and cellular thermal shift assays, has confirmed that Gastrodin can directly bind to the P2RY12 receptor and inhibit the activation of its downstream RhoA/ROCK signaling pathway. However, this study still has certain limitations. This study mainly relied on *in vitro* cell models. Although this model has good experimental controllability, it still cannot fully simulate the complex microenvironment in the process of epilepsy occurrence *in vivo*, especially the complex neuro-immune interaction network. Subsequent studies need to use epilepsy mouse models with microglia-specific *P2RY12* knockout to further verify the target specificity and neuroprotective effect of Gastrodin *in vivo*. In addition, although we have confirmed the direct binding effect of Gastrodin and P2RY12, whether Gastrodin has cross-interactions with other purinergic receptors, and issues such as its *in vivo* pharmacokinetic characteristics, blood-brain barrier penetration efficiency, and long-term drug administration safety still need to be further studied.

In summary, this study systematically reveals that Gastrodin alleviates epilepsy-associated neuroinflammation and neuronal damage by directly targeting P2RY12 and inhibiting downstream RhoA/ROCK-mediated microglial migration, thereby providing a potential therapeutic strategy for epilepsy intervention.

## Supporting information

**S1 File.** S1 Fig. Validation of the knockdown model for P2RY12. a. qPCR assay was used to detect the knockdown of *P2RY12* mRNA in HMC3 microglial cells, with n = 3. b-c. Western blot assay was employed to verify the downregulation of P2RY12 protein expression in HMC3 microglial cells, with n = 3. Supplementary Table 1. Primer sequence table. The sequences of qPCR primers used in this study.
(DOCX)

**S1 Raw image. S1-raw-images file: Presents the complete membrane results obtained through the protein blotting experiment.** The figure captions in the document, in order from top to bottom, are: a-b Correspond to Figure 4a in the original text; c-d Correspond to Figure 4b in the original text. a-b Correspond to Figure 4c in the original text. a-c Correspond to Figure 7a in the original text; d Correspond to Supplementary Figure 1b in the supplementary material text.
(PDF)

## Author contributions

**Data curation:** Hailong Huang.

**Formal analysis:** Ping Liu.

**Methodology:** Aiyuan Cai, Zilong Li, Ruizhong Zhang, Yuanhong Lin.

**Supervision:** Ran Liu.

**Validation:** Jing Xiao.

**Writing – original draft:** Aiyuan Cai.

**Writing – review & editing:** Qingpeng Hu, Haixia Wu.

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
