## [Decision Letter · Decision Letter 0]

18 Feb 2026

Gastrodia elata Alleviates Neuronal Damage in Chronic Epilepsy by Targeting P2RY12 Receptor to Inhibit Excessive Activation of Microglia

PLOS One

Dear Dr. Hu,

Thank you for submitting your manuscript to PLOS ONE. After careful consideration, we feel that it has merit but does not fully meet PLOS ONE’s publication criteria as it currently stands. Therefore, we invite you to submit a revised version of the manuscript that addresses the points raised during the review process.

Please submit your revised manuscript by Apr 03 2026 11:59PM. If you will need more time than this to complete your revisions, please reply to this message or contact the journal office at plosone@plos.org . . Please include the following items when submitting your revised manuscript:

We look forward to receiving your revised manuscript.

Kind regards,

Xiaona Wang, Ph.D

Academic Editor

PLOS One

Journal Requirements:

“This research is funded by Protective effect of SAHA on brain injury in developing epileptic rats and regulation of histone acetylation of TLR4 gene (2023JJ30532); The anti-epileptic mechanism of xyloketal B was explored based on network pharmacology from SIRT1/NF-κB/GMD mediated astrocytic pyroptosis, Excellent Project of the Shenzhen Longhua District Science and Technology Innovation Bureau (2025012); Research on improving the teaching effect of pediatric neurological diseases in Longhua District People's Hospital, teaching reform of Shenzhen Longhua District People's Hospital (2024251).”

4. We note that your Data Availability Statement is currently as follows: All relevant data are within the manuscript and in Supporting Information files.

6. Please remove your figures from within your manuscript file, leaving only the individual TIFF/EPS image files, uploaded separately. These will be automatically included in the reviewers’ PDF.

7. PLOS ONE now requires that authors provide the original uncropped and unadjusted images underlying all blot or gel results reported in a submission’s figures or Supporting Information files. This policy and the journal’s other requirements for blot/gel reporting and figure preparation are described in detail at https://journals.plos.org/plosone/s/figures#loc-blot-and-gel-reporting-requirements and https://journals.plos.org/plosone/s/figures#loc-preparing-figures-from-image-files. When you submit your revised manuscript, please ensure that your figures adhere fully to these guidelines and provide the original underlying images for all blot or gel data reported in your submission. See the following link for instructions on providing the original image data: https://journals.plos.org/plosone/s/figures#loc-original-images-for-blots-and-gels.

Reviewers' comments:

Reviewer's Responses to Questions

**Comments to the Author**

1. Is the manuscript technically sound, and do the data support the conclusions?

Reviewer #1: Yes

Reviewer #2: Yes

2. Has the statistical analysis been performed appropriately and rigorously?

Reviewer #1: I Don't Know

Reviewer #2: Yes

3. Have the authors made all data underlying the findings in their manuscript fully available?

Reviewer #1: Yes

Reviewer #2: Yes

4. Is the manuscript presented in an intelligible fashion and written in standard English?

Reviewer #1: Yes

Reviewer #2: Yes

Reviewer #1: The description of statistical methods lacks sufficient detail. Please specify the exact statistical test used for each comparison (e.g., unpaired two-tailed t-test, one-way ANOVA with Tukey’s post hoc test), and clarify whether assumptions of normality and homogeneity of variance were assessed.

Reviewer #2: The article systematically investigated the effect and mechanism of Gastrodin, one of the main components of Gastrodia elata, on P2RY12 in a cell model of epilepsy. Experimental results indicated that Gastrodin suppressed microglial migration and F-actin remodeling, reduced TNF-α and IL-1β release, prevented neuronal calcium overload and apoptosis in epilepsy model cells, and thereby mitigated neuroinflammation and neuronal damage. The manuscript presents sound scientific research with significant reference value, but requires further improvement.

1. The title of the article requires improvement. The article primarily conducts research on the effect and mechanism of Gastrodin on P2RY12 in the epilepsy cell model. Therefore, the terms "Gastrodia elata" and "in Chronic Epilepsy" in the title are not sufficiently accurate.

2. In line 348~349, the contents of Figure 1d should be explained in detail. In addition, it is suggested that Figure 1d change the colors to more clearly represent the intersection of the three sets. The same modification applies to Figure 2e, as the black is too dark and impairs visual clarity.

3. The title of Figure 8, “Schematic diagram of the P2RY12-mediated signaling pathway in microglia in the epileptic state”, fails to reflect the regulation mechanism of Gastrodin, and need to be improved.

4. The paragraph (lines 682–693) lacks objectivity and rigor; it should rely on scientific facts, not subjective speculation. This also applies to the last sentence of the concluding paragraph.

5. Some symbols and formats are misused in this article, such as CO2 and so on. Moreover, the capitalization of the initial letter of Gastrodin is not uniform throughout the text. Please check and revise thoroughly.

**Do you want your identity to be public for this peer review?**  For information about this choice, including consent withdrawal, please see our  For information about this choice, including consent withdrawal, please see our Privacy Policy .

Reviewer #1: No

Reviewer #2: No

---

## [Author Response · Author response to Decision Letter 1]

20 Mar 2026

Dear Xiaona Wang, Ph.D,

Thank you very much for considering our manuscript entitled "Gastrodin Alleviates Neuronal Damage in Epileptic Cell Models by Targeting P2RY12 to Inhibit Microglial Hyperactivation" (Manuscript number: PONE-D-25-64129) for publication in PLOS One. We are grateful to both the reviewers and the editor for their constructive comments and suggestions, which have greatly helped us improve the quality of the paper. We have carefully considered all the comments and have made the appropriate revisions to the manuscript. Below, we provide detailed responses to each of the points raised:

Response to Academic Editor

The funding information has been completely removed from the main text. We confirm that all funding‑related content has been deleted from the manuscript, and the funding declaration is only provided via the online submission system. The relevant statement has been added and updated in the cover letter accordingly.

Figures and blots/gels: All figures have been removed from the main text file and uploaded separately as TIFF files. All original uncropped, unadjusted blot/gel images are provided in the Supporting Information.

Response to Reviewer #1

Comment: The description of statistical methods lacks sufficient detail. Please specify the exact statistical test used for each comparison, and clarify whether assumptions of normality and homogeneity of variance were assessed.

Response:We sincerely appreciate the valuable comments and suggestions you have provided on our manuscript. We deeply apologize for the inadequate description of the statistical methods. We have thoroughly revised the section "2.12 Data Statistical Methods" to clearly state the following: the Shapiro-Wilk test is employed for normality testing, the Levene test is used for homogeneity of variance testing, an unpaired two-tailed t-test is applied for comparisons between two groups, one-way analysis of variance followed by Tukey's test is conducted for multiple-group comparisons, and for non-parametric data, the Kruskal-Wallis test combined with Dunn's test is utilized. The relevant modifications have been made in lines 316 to 324.

Response to Reviewer #2

Comment 1: The title is inaccurate. “Gastrodia elata” and “chronic epilepsy” are not sufficiently accurate.

Response:We are extremely grateful for your strict comments on the inaccuracy of the original title. In response to your suggestions, we have carefully revised the title to precisely define the research subject, experimental system and core mechanism. The revised title is as follows: Gastrodin Alleviates Neuronal Damage in Epileptic Cell Models by Targeting P2RY12 to Inhibit Microglial Hyperactivation.

Comment 2: Lines 348–349: Explain Figure 1d in detail; change colors of Figure 1d and Figure 2e for clarity.

Response:We sincerely thank you for your constructive suggestions on improving the presentation of the charts and graphs. We have supplemented a detailed description and interpretation of Figure 1d in lines 345 to 351 of the Results section. Additionally, the color schemes of Figure 1d (a Venn diagram) and Figure 2e have been adjusted to high-contrast light tones to enhance visual clarity and readability.

Comment 3: Figure 8 title fails to reflect the regulatory mechanism of Gastrodin; needs improvement.

Response:We sincerely appreciate your constructive feedback. To enhance clarity and accuracy, we have revised the title of Figure 8 to: "The mechanism by which gastrodin targets P2RY12 to inhibit the RhoA/ROCK pathway and microglial migration in an epilepsy model."

Comment 4: Paragraph lines 682–693 and the last sentence of the conclusion lack objectivity and rigor.

Response: We sincerely appreciate this constructive suggestion. We have carefully revised the paragraph spanning lines 682 to 693, as well as the last sentence of the conclusion section, to eliminate speculative and exaggerated expressions. All descriptions are now strictly based on the data and results obtained in this study, and the scientific objectivity and rationality of the conclusions have been significantly enhanced. Now, all conclusions are fully supported by our experimental evidence. The relevant modifications have been made in lines 677 to 693.

Comment 5: Errors in symbols (e.g., CO₂) and inconsistent capitalization of Gastrodin.

Response: We sincerely appreciate your meticulous corrections. We have carefully examined and revised the entire manuscript: the incorrect expression "CO2" has been uniformly corrected to "CO₂", and the capitalization of "Gastrodin" has been consistently maintained throughout the text to ensure formatting uniformity. All symbol and capitalization errors have been duly corrected.

We hope that the revisions and clarifications we have made will satisfactorily address the concerns raised by the reviewers. We would be more than willing to make further adjustments if necessary.

Thank you again for your consideration and for the opportunity to submit our work to PLOS ONE.

Sincerely,

Qingpeng Hu: huqingpeng163@126.com

Haixia Wu: 13923430437@163.com

---

## [Editor Report · Decision Letter 1]

25 Mar 2026

Gastrodin Alleviates Neuronal Damage in Epileptic Cell Models by Targeting P2RY12 to Inhibit Microglial Hyperactivation

PONE-D-25-64129R1

Dear Dr. Hu,

We’re pleased to inform you that your manuscript has been judged scientifically suitable for publication and will be formally accepted for publication once it meets all outstanding technical requirements.

Kind regards,

Xiaona Wang, Ph.D

Academic Editor

PLOS One

---

## [Editor Report · Acceptance letter]

PONE-D-25-64129R1

PLOS One

Dear Dr. Hu,

I'm pleased to inform you that your manuscript has been deemed suitable for publication in PLOS One. Congratulations! Your manuscript is now being handed over to our production team.

Kind regards,

on behalf of

Associate Professor Xiaona Wang

Academic Editor

PLOS One